# A SIMPLE CONTRASTIVE LEARNING OBJECTIVE FOR ALLEVIATING NEURAL TEXT DEGENERATION

## ABSTRACT

The cross-entropy objective has proved to be an all-purpose training objective for autoregressive language models (LMs). However, without distinguishing problematic tokens, LMs trained using cross-entropy exhibit text degeneration problems. To address this, unlikelihood training has been proposed to reduce the probability of unlikely tokens predicted by LMs. But unlikelihood does not explicitly consider the relationship between the label tokens and unlikely token candidates, thus showing marginal improvements in degeneration. We propose a new *contrastive token* learning objective that inherits the advantages of cross-entropy and unlikelihood training and avoids their limitations. The key idea is to teach a LM to generate high probabilities for label tokens and low probabilities for negative candidates. Comprehensive experiments on language modeling and open-domain dialogue generation tasks show that the proposed contrastive token objective yields much less repetitive texts, with a higher generation quality than baseline approaches, achieving the new state-of-the-art performance on text degeneration.

## 1 INTRODUCTION

Autoregressive language models (LMs), such as OpenAI GPT-3 (Brown et al., 2020), have achieved impressive results on various natural language processing tasks. The goal of training LMs is to learn the true distribution of a text corpus, and this is usually achieved through next word prediction. Specifically, a standard approach to training LMs is to minimize the cross-entropy loss between the true distribution and the model prediction. Unfortunately, LMs trained using the cross-entropy objective have been observed to exhibit text degeneration problems, where token, phrase, and sentence level repetition is a common symptom (Holtzman et al., 2020; Welleck et al., 2020; Jiang et al., 2020). Such repeated texts differ markedly from those generated by humans.[1] To analyze the reasons for degeneration, our work views the vocabulary of LMs as being composed of three sets of tokens at each time step, i.e., positive tokens (label tokens), negative tokens (incorrectly repeating tokens), and irrelevant tokens (all the others). Based on this taxonomy, we stress that cross-entropy is in fact a contrastive learning objective that contrasts positive tokens with all negative and irrelevant tokens. While it is necessary for LMs to learn how to rank positive tokens higher than other tokens in the predicted distribution, negative tokens are treated equally as irrelevant tokens (whose amount is usually much larger) by the cross-entropy objective. As a consequence, negative tokens may not be suppressed hard enough.

To address the above issue, Welleck et al. (2020) have proposed *unlikelihood training* to penalize certain negative tokens, i.e., tokens that are incorrectly repeated. The key idea behind unlikelihood training is to lower the probability of negative tokens assigned by LMs. Despite its success, the unlikelihood objective does not explicitly consider the relationship between positive and negative tokens, which causes it to have indefinite effects on suppressing negative tokens. Unlikelihood training also unintentionally boosts the probability of other irrelevant tokens. Moreover, all previous context tokens are used as negative candidates per prediction step, which not only introduces a considerable amount of noise, but also results in sub-optimal repetition reduction, thus affecting the final generation performance.

---

[1]Readers are referrred to Table 4 for some concrete examples. The degeneration problem even exists in large-scale state-of-the-art pre-trained language models such as GPT-3 (Ouyang et al., 2022).

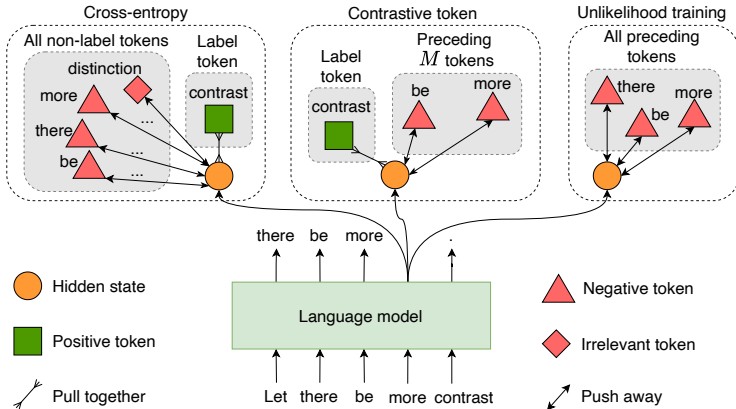

Figure 1: Illustrating the differences between our proposed contrastive token learning, unlikelihood training, and the cross-entropy objective for LMs. For contrastive token learning, we use the label token as the positive token and the preceding $M$ tokens as the negative tokens at each decoding step.

Table 1: The influence comparison of different learning objectives over the positive (label), negative (incorrectly repeating), and irrelevant tokens (all the others) for the LMs.

| Loss | Relevant tokens | | Irrelevant tokens | Contrast |
|---|---|---|---|---|
| | Positive | Negative | | |
| Cross-entropy (CE) | Promote | Suppress | Suppress | Yes |
| Unlikelihood training (UL) | Promote | Suppress/Promote | Promote | No |
| Contrastive token (CT) | Promote | Suppress | Unchanged | Yes |

In this paper, we introduce a simple yet effective *contrastive token learning* (CT for short) objective that integrates the best of cross-entropy and unlikelihood training, penalizing negative tokens by contrasting them directly with positive tokens. The commonalities and differences between cross-entropy, unlikelihood training, and CT are illustrated in Figure 1. Briefly, (i) without distinguishing between negative and irrelevant tokens, cross-entropy cannot effectively suppress negative tokens; (ii) due to the lack of contrast between negative and positive tokens, it is ineffective for unlikelihood training to penalize negative tokens; and (iii) through its direct contrast between positive and negative tokens, CT is more focused in learning the differences between them, i.e., explicitly teaching the LM to assign negative tokens with a lower probability than positive tokens. In this work, we combine the CT and cross-entropy objectives to train LMs, where cross-entropy operates on the label tokens so that they are assigned the highest probability, and CT effectively suppresses negative tokens from being generated.

We perform evaluations on the tasks of language modeling (decoder-only model) and open-domain dialogue generation (encoder-decoder model). Our empirical evidence demonstrates that LMs trained with the proposed CT objective can generate much less repetitive texts and achieve superior text generation performance under both automatic and human evaluations. CT has a minor negative influence on the perplexity of LMs, but thanks to the reduced repetition rates, in our case studies we observe substantial improvements regarding the quality of generated text.

## 2 BACKGROUND

LMs aim to learn the true distribution over variable-length text sequences in a text corpus $X = (x_1, x_2, \ldots, x_{|X|})$ with $|X|$ tokens. A popular approach to this task is next word prediction, i.e., predicting a distribution over the next word following a given context. To train such a language model, cross-entropy and unlikelihood training are two representative objectives, which we will first review in this section. We then provide an analysis of the text degeneration problem.

## 2.1 Cross entropy

A standard approach to training a LM is to minimize the expected cross-entropy loss between the true distribution and the model prediction (Yang et al., 2019a). Specifically, the cross-entropy loss for each time step $t$ is defined as:

$$\mathcal{L}_{CE}^t = -\log p(x_t|x_{<t}) \tag{1}$$

$$= -\log \frac{\exp(h_t^T W_{x_t})}{\sum_{\hat{x}_t \in V} \exp(h_t^T W_{\hat{x}_t})} \tag{2}$$

$$= \log \left(1 + \sum_{\hat{x}_t \in V, \hat{x}_t \neq x_t} \exp(h_t^T W_{\hat{x}_t} - h_t^T W_{x_t})\right), \tag{3}$$

where $h_t$ is the model hidden state at time $t$, $W$ is the embedding matrix, and $W_{x_t}$ denotes the embedding of token $x_t$. Through the transformations from Eq. (1)–(3), we can see that Eq. (3) is similar to the $N$-pair contrastive loss (Sohn, 2016) for visual object recognition. In other words, cross-entropy effectively trains LMs to contrast the label tokens (positive examples) $x_t$ with all the other non-label tokens (negative and irrelevant examples) $\hat{x}_t \in V, \hat{x}_t \neq x_t$ in the whole vocabulary.

## 2.2 Unlikelihood training

To address the repetition issue of cross-entropy, Welleck et al. (2020) proposed unlikelihood training to penalize negative tokens (UL-T). The unlikelihood loss for time step $t$ is defined as:

$$\mathcal{L}_{UL}^t = -\sum_{x_t^- \in C^t} \log(1 - p(x_t^-|x_{<t})), \tag{4}$$

where $C^t = \{x_1, \ldots, x_{t-1}\} \backslash \{x_t\}$ is the set of negative tokens at time $t$, i.e., all previous context tokens. In this paper, we refer to this set of negative tokens as the *preceding tokens set*. As we will see in §2.3, UL-T does not work well as it can increase the probability of irrelevant tokens. Welleck et al. (2020) have also proposed a more effective *sequence-level unlikelihood objective* (UL-S) that uses unlikelihood on generated continuations during training time. We omit the details here as our proposed CT is more closely related to UL-T, but we compare CT to UL-S in our experiments.

## 2.3 Discussion

The main difference between Eq. (3) and the $N$-pair contrastive loss is that, in Eq. (3), negative and irrelevant tokens are treated equally by cross-entropy.[2] These negative tokens need to be penalized harder than irrelevant tokens, otherwise, negative tokens may be incorrectly repeated in later time steps. We believe this to be the reason why LMs trained by cross-entropy have high repetition rates.

Although UL-T penalizes negative tokens, it is not effective enough. As can be seen from Table 1, the reasons are twofold. First, each negative token is not definitely penalized because it depends on other negative tokens, which can be seen from the gradient analysis of UL-T (Eq. (11) in Appendix C). Second, the formulation of UL-T unintentionally boosts the probability of other irrelevant tokens and may make them surface as repeated tokens. We detail this analysis in §3.3.

## 3 Method

To address the issues discussed above and inherit the advantages of cross-entropy and unlikelihood training, in this section, we present a novel contrastive token learning (CT) objective. We first define the CT loss for each time step. Then we introduce a negative token selection strategy. Finally, we discuss the relationships among CT, cross-entropy and unlikelihood training.

## 3.1 Contrastive token learning

The key idea of CT is to promote positive (label) tokens in the ranking at each step, while lowering negative (incorrectly repeating) tokens, and leave other irrelevant tokens unchanged. To this end, we

---

[2] Albeit with different strengths, as seen in Eq. (10) in Appendix C.

formulate the CT loss for step $t$ as:

$$\mathcal{L}_{CT}^t = \log \left( 1 + \sum_{x_t^- \in S_N^t} \exp(h_t^T W_{x_t^-} - h_t^T W_{x_t}) \right), \tag{5}$$

where $S_N^t$ is the negative token set and $x_t$ is the positive token (i.e., label token) at time $t$. We detail the token selection mechanism of $S_N^t$ below. Comparing Eq.(5) to Eq. (4), we see that UL only considers the probabilities of negative tokens, while CT directly contrasts negative with positive tokens. During training, we combine the CT loss with the cross-entropy loss for each time step:

$$\mathcal{L}^t = \mathcal{L}_{CE}^t + \mathcal{L}_{CT}^t. \tag{6}$$

$\mathcal{L}_{CE}^t$ trains LMs to assign the highest probabilities to label tokens. While on the other hand, $\mathcal{L}_{CT}^t$ focuses on contrasting positive tokens and negative tokens, so that the LMs can learn to effectively rank negative tokens lower than their positive counterparts.

## 3.2 NEGATIVE TOKEN SELECTION STRATEGY

Following (Welleck et al., 2020), we also select negative tokens from the preceding tokens. However, using all preceding tokens (as done in (Welleck et al., 2020)) introduces too many irrelevant tokens, especially in later time steps of a sequence. Hence, we instead propose to use the *preceding $M$ tokens set* to decide the negative tokens, with $M$ being a hyper-parameter. The set $S_N^t$ is defined as:

$$S_N^t = \{x_{t-M}, \ldots, x_{t-1}\} \backslash \{x_t\}. \tag{7}$$

Another difference with the *preceding tokens set* (Welleck et al., 2020) is that, $S_N^t$ is a *multiset* that does not remove redundant occurrences. Intuitively, minimizing the CT loss with the *preceding $M$ tokens set* makes more frequently repeated tokens less likely to be predicted.

## 3.3 GRADIENT ANALYSIS

To see how loss functions influence the positive, negative and irrelevant tokens during training, we derive the gradient functions of each loss function with respect to these tokens in Appendix C. Table 1 is an intuitive summary of the influences, from which one can observe that: (i) Cross-entropy trains to promote label tokens in rankings at each time-step, while suppressing all the other tokens including negative and irrelevant tokens. (ii) It cannot be decided for unlikelihood training whether the negative tokens are promoted or suppressed by the gradient function (cf. Eq. (11) in Appendix C, the valid region for the corresponding gradient function contains both positive and negative values), and irrelevant tokens are promoted, both of which are problematic. (iii) With contrastive token learning, CT promotes positive tokens and suppresses negative tokens, and it is the only objective that does not affect irrelevant tokens (cf. the gradient functions in Appendix C).

When using CT together with CE, as we do for our final loss function, negatives are suppressed both in CT and in CE, while irrelevant tokens are only suppressed in CE. Therefore, our CT objective is able to better restrain incorrectly repeated tokens.

## 4 RELATED WORK

We review two lines of related work, i.e., neural text degeneration and contrastive learning.

**Neural text degeneration.** With large-scale pre-training, state-of-the-art neural LMs are able to generate human-like texts (Brown et al., 2020; Yang et al., 2019a). However, they suffer from the *text degeneration problem*, where model-generated texts are dull and repetitive (Jiang & de Rijke, 2018; Holtzman et al., 2020; Welleck et al., 2020). The text degeneration problem is especially serious with open-ended generation tasks, such as dialogue generation (See et al., 2019; Jiang et al., 2020) and language modeling (Holtzman et al., 2020; Welleck et al., 2020). Some decoding approaches have been proposed to address this problem, by introducing randomness (Fan et al., 2018; Holtzman et al., 2020) or disparity (See et al., 2019; Su et al., 2022) at inference time. Some other work suggests that the degeneration problem is caused by defects of the likelihood training objective, and

improved training objectives have been proposed (Jiang et al., 2019; Welleck et al., 2020; Su et al., 2022). ScaleGrad Lin et al. (2021) encourages the LMs to generate novel tokens, but the selection of such tokens can be too open.

Our proposed contrastive token learning approach belongs to the training objective family. Compared to unlikelihood training (Welleck et al., 2020), we address the suppression of repetitive tokens by contrasting them with positive tokens.

**Contrastive learning.** In computer vision, contrastive learning has been widely employed to learn representations (Sohn, 2016; Chen et al., 2020; Khosla et al., 2020). Noise-contrastive estimation (Gutmann & Hyvärinen, 2010) has been proved successful for training word embeddings (Mikolov et al., 2013). In recent years, contrastive learning has gained more attention in the area of natural language processing too. Most work builds contrast at the sequence or document level by corrupting the ground truth sequence (Yang et al., 2019b; Clark et al., 2020; Lee et al., 2021; Meng et al., 2021) or mining positive/negative samples (Nguyen & Luu, 2021; Pan et al., 2021).

Existing token-level contrastive learning frameworks contrast model representations from different positions (Zhang et al., 2021; Su et al., 2022). Differently, we contrast word embeddings while using the hidden representations as anchor points similar to the triplet contrastive loss (Schroff et al., 2015). Our formulation effectively contrasts logits output by the model for positive and negative tokens, thus it is more direct than unlikelihood training on addressing the repetitive degeneration problem. To the best of our knowledge, our proposed CT is the first to use token embeddings as positive/negative examples in a contrastive framework for the text degeneration problem.

## 5 EXPERIMENTAL SETUP

We compare CT with baseline approaches on the language modeling and open-domain dialogue generation task (using an encoder-decoder model). Since our experimental results on the dialogue task show a similar pattern as on the language modeling task, we will focus on the language modeling task in the body of the paper and postpone the setup and analyses of the dialogue task to Appendix H.

**Baselines and implementation.** We implement several state-of-the-art baselines and use them with GPT-2 (Radford et al., 2019): (i) For decoding-based methods, we consider: banning 3-grams (Roller et al., 2021), top-$k$ sampling (Fan et al., 2018), nucleus sampling (Holtzman et al., 2020) and contrastive search (SimCTG-CS) (Su et al., 2022); and (ii) learning-based methods: unlikelihood training (Welleck et al., 2020), SimCTG (Su et al., 2022), and noise-contrastive estimation (NCE; detailed in Appendix B) (Gutmann & Hyvärinen, 2010). We also consider model trained using CE as a baseline. More details can be found in Appendix D.

**Dataset, training and inference details.** At training time, we fine-tune GPT-2 small on the widely-used Wikitext-103 dataset (Merity et al., 2017) with each learning-based approach (including the CE baseline) for 50K steps with 3K warm-up steps. As suggested in (Welleck et al., 2020), for sequence-level unlikelihood training, we first fine-tune the language model using UL-T for 48.5K steps, and then switch to the UL-S objective for another 1.5K steps, resulting in UL-TS. Best model checkpoints for each task are selected according to the lowest validation CE loss with an evaluation interval of 1K training steps. We use trunks of 512 tokens, and a training batch size of 4. All models are trained using the Adam optimizer (Kingma & Ba, 2014) with a learning rate of 1e-5. For UL-TS, we had to use a smaller learning rate of 1e-6, otherwise the generated texts contain massive ungrammatical repetitions (continuous token repetitions, as can be seen in Table 5 of Appendix E).

At inference time, we compare the performance of each approach using both greedy search and beam search. Following the best settings reported on this task (Welleck et al., 2020), we use $k = 50$ for top-$k$ sampling, and $p = 0.9$ for nucleus sampling. We follow Welleck et al. (2020) to use 50 tokens as the input prefix and let the model generate 100 tokens as a continuation.

**Evaluation metrics.** We measure the perplexity (`ppl`) of different approaches. For measuring generative repetition, we follow Welleck et al. (2020) to use 1-gram to 4-gram repetition rates (`rep-1` – `rep-4`), which are defined as the number of repeated $n$-grams divided by the total number of generated $n$-grams in each sequence, micro-averaged over the whole dataset. We also report the generation diversity at the dataset level, which is measured by distinct 1-gram rates (`dist-1`) (Li et al., 2016) and unique 1-gram counts (`uniq-1`). We adopt human evaluation for measuring the

Table 2: Results on the test set of Wikitext-103 for the language modeling task. ↑/↓ arrows denote whether higher or lower is better for a metric. The best result for either type of approach (decoding-based vs. learning-based) under each metric is highlighted in **bold face**. ‡ Does not count as the best. † For this experiment, we use a beam size of 5 as suggested in its original paper (Su et al., 2022).

| | | ppl↓ | ppl-s↓ | search | rep-1↓ | rep-2↓ | rep-3↓ | rep-4↓ | dist-1↑ | uniq-1↑ |
|---|---|---|---|---|---|---|---|---|---|---|
| | GPT-2 | 18.01 | 25.95 | greedy | 71.03 | 60.12 | 54.77 | 50.93 | 1.15 | 12787 |
| | | | | beam | 77.02 | 69.70 | 65.49 | 61.69 | 1.12 | 12545 |
| *decoding-based* | 3-gram ban | 18.01 | 25.95 | greedy | 50.09 | 18.31 | 0.00‡ | 0.00‡ | 1.52 | 16940 |
| | | | | beam | 40.91 | 10.40 | 0.00‡ | 0.00‡ | 1.35 | 15114 |
| | Top-$k$ | 18.01 | 25.95 | greedy | **34.80** | **9.38** | **3.86** | **1.73** | **2.23** | **24840** |
| | | | | beam | 73.47 | 64.38 | 59.31 | 54.88 | 1.19 | 13280 |
| | Nucleus | 18.01 | 25.95 | greedy | 38.41 | 12.10 | 5.50 | 2.78 | 2.06 | 23038 |
| | | | | beam | 74.28 | 65.70 | 60.86 | 56.58 | 1.17 | 13004 |
| | SimCTG-CS | 18.12 | 26.10 | greedy | 70.23 | 58.92 | 53.44 | 49.54 | 1.17 | 13005 |
| | | | | beam† | **31.93** | **6.52** | **2.23** | **0.94** | **1.77** | **19746** |
| *learning-based* | SimCTG | **18.12** | **26.10** | greedy | 70.23 | 58.92 | 53.44 | 49.54 | 1.17 | 13005 |
| | | | | beam | 75.87 | 68.02 | 63.54 | 59.52 | 1.15 | 12835 |
| | NCE | 18.60 | 32.88 | greedy | 57.23 | 41.59 | 35.50 | 31.75 | 1.32 | 14774 |
| | | | | beam | 56.02 | 40.99 | 34.73 | 30.48 | 1.28 | 14322 |
| | UL-T | 18.93 | 26.63 | greedy | 60.91 | 45.15 | 38.31 | 33.90 | 1.26 | 14071 |
| | | | | beam | 67.39 | 55.95 | 49.85 | 44.78 | 1.15 | 12874 |
| | UL-TS | 18.88 | 27.41 | greedy | 51.98 | 29.17 | 19.71 | 14.42 | 1.29 | 14378 |
| | | | | beam | 45.81 | 23.96 | 15.60 | 10.41 | 1.27 | 14141 |
| | CT | 18.67 | 52.77 | greedy | **26.74** | **8.23** | **3.73** | **1.52** | **1.93** | **21562** |
| | | | | beam | **31.13** | **13.66** | **9.28** | **7.00** | **1.61** | **18016** |
| | Human | – | – | – | 29.92 | 7.25 | 2.81 | 1.14 | 3.41 | 19034 |

quality of model generated texts. We randomly select 100 prefixes from the test set of Wikitext-103, and compare the continuations generated using CT with those by the best-performing baselines according to the automatic evaluation results. Since it does not make much sense to compare continuations with either side having excessive repetitions, we filter out such pairs using a threshold of `rep-4` $\leq 0.05$ to make the comparisons more competitive. Then we display the prefix and two continuations from different systems (side-by-side, in a random order) to three crowd workers and ask them to select the winner in terms of repetition, coherence, fluency, and overall quality. Ties are allowed for all aspects. We use majority voting to decide the final winner. Details about our question form design and the instructions to crowd workers can be found in Appendix F.

## 6 EVALUATION RESULTS

In this section, we discuss how CT compares to SOTA methods under both the automatic and human evaluations, as well as showing some visualization analysis on its generation pattern.

### 6.1 BASELINE COMPARISON

The performance comparisons between our CT and the baselines on the language modeling task are shown in Table 2. For understanding the model performance relative to human, we also calculate these metrics on human-created text. The `ppl` metric is for 512-token sequences to comply with the training sequence length. To be comparable to existing work (Welleck et al., 2020; Su et al., 2022), we also report `ppl-s` for short sequences of 50 tokens. We use a sequence length of 150 tokens and $M = 30$ as the negative window size for CT. Justifications for such hyper-parameter selections can be found in Appendix E.2.

**CT compared to learning-based approaches.** One can observe that CT performs the best and its performance is very close to humans according to `rep-*` rates and unique token counts (`uniq-1`)

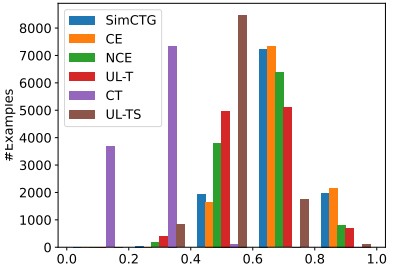 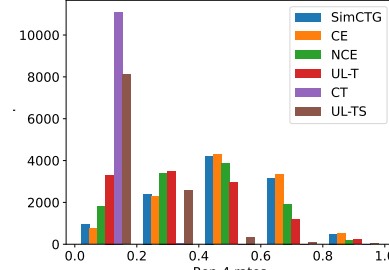

Figure 2: Histograms for `rep-1` (left) and `rep-4` (right) rates of each method, on the Wikitext-103 test set.

when using greedy search. However, we can still *not* conclude that the repetition problem is solved, because when looking at specific cases, models trained by CT still occasionally generate texts with excessive repetitions, although much rarer than baseline methods. To see how each method performs at every repetition level, we group the `rep-1` and `rep-4` rates of model-generated texts in to 5 bins, and plot their histograms in Figure 2, from which we can see that CT generates substantially less degenerated continuations (with `rep-1`$\geq$ 0.4 and `rep-4`$\geq$ 0.2). For UL-TS, we were able to achieve lower repetition rates with a larger learning rate of 1e-5 during training. However, the trained LM often generates ungrammatical repetitions. This problem does not exist with CT. The comparisons are shown in Table 5 in Appendix E, and in §6.3 we show that this is caused by UL-TS being uncertain about its predictions at later time steps.

The diversity improvements brought by CT are the largest among all learning-based methods, especially when using greedy search. CT increases the second highest `uniq-1` count (NCE) by 46%. When compared to UL-T, one can see that utilizing the contrast between positive and negative tokens works better than solely penalizing negative tokens. Comparing SimCTG to the CE baseline, one can observe that the contrastive objective of SimCTG itself has very limited effect on reducing repetition, which is also mentioned in the original paper (Su et al., 2022). This is because SimCTG contrasts hidden states of positive (current step) and negative (other steps) tokens, but it does not consider the influence of token embeddings on the repetition problem, as done in CT.

The `ppl` increase brought by CT is minor, with 0.66 points. When calculated on short sequences, due to the length mismatch of training and test sequences, `ppl-s` scores are higher than `ppl` for all approaches. Among them, contrastive objectives (NCE and CT) have larger `ppl-s` increases than other methods. Although CT has the highest increase on `ppl-s`, our case study (Table 4) shows that the generation quality of CT is not harmed, but on the contrary is improved due to the lower repetition and higher diversity of the generated texts.

**CT compared to decoding-based approaches.** Although CT is a learning-based method, we still compare it against decoding approaches for a more comprehensive understanding of its performance. When greedy search is used, CT outperforms the best decoding method (Top-$k$) in terms of `rep-*` rates, which again proves the effectiveness of contrastive learning. When using beam search, all but SimCTG-CS perform significantly worse than CT, both in terms of repetition rates and diversity. SimCTG-CS is effective at reducing repetition as it explicitly requires a disparity among different time steps at inference time. This can harm the generation quality, especially the coherence and fluency, as we see in §6.2. It is also worth noting that SimCTG-CS only works together with its SimCTG training objective and with beam search (Su et al., 2022). In summary, one can see that the repetition problem can be better addressed from the model learning perspective, in which case a simple greedy decoding strategy suffices.

## 6.2 HUMAN EVALUATION

Human evaluation results are shown in Table 3. Regarding the overall quality, CT performs significantly better than Top-$k$ and SimCTG-CS, two decoding based approaches. Instead of purely learning generation policies from data, decoding approaches exert heuristics at inference time, which may prevent the language model from performing naturally. This explains the worse performance of

Table 3: Win/lose rates (%) of CT compared to baselines under human evalutaions. For a competitive comparison, we filtered out highly repetitive examples of either model in the pair. * indicates statistical significance as determined with a sign test ($p < 0.05$).

| Comparison | Overall | | Repetition | | Coherence | | Fluency | |
|---|---|---|---|---|---|---|---|---|
| | Win | Lose | Win | Lose | Win | Lose | Win | Lose |
| CT vs Top-$k$ | 58* | 36 | 40* | 23 | 56* | 36 | 45 | 36 |
| CT vs SimCTG-CS | 55* | 35 | 46* | 18 | 52 | 36 | 54* | 28 |
| CT vs UL-TS | 48 | 43 | 43 | 28 | 39 | 45 | 47 | 38 |
| CT vs Human | 27 | 67* | 30 | 35 | 23 | 67* | 27 | 57* |

decoding approaches on coherence and fluency. CT performs generally better than UL-TS except on coherence, but none of these differences are statistically significant. This suggests that CT has a similar generation quality as UL-TS on low-repetitive examples, but CT has much lower repetition rates as reported in Table 2. This result is expected, as both CT and UL-TS are learning-based approaches for training data-driven models, and on normal cases such as low-repetitive generations, they should perform similarly. Compared to human performance, there is still a large margin for machine learning models before they have a comparable performance on the language modeling task. Although CT performs on par with humans regarding repetition, its generations are far less coherent and fluent than those of humans. This may be mitigated by using larger models such as GPT-2 large or GPT-3. However, we could not perform such experiments due to a lack of computational resources.

### 6.3 VISUALIZATION ANALYSIS OF THE GENERATION PROBABILITY

We also conduct analysis to understand the predicted probability of model-generated tokens at inference time. As shown in Figure 3, diagonal cells represent the probability of generated tokens at the corresponding time steps; off-diagonal cells represent the probability of context tokens. The plots are averaged over 10 random instances from the test set of Wikitext-103.

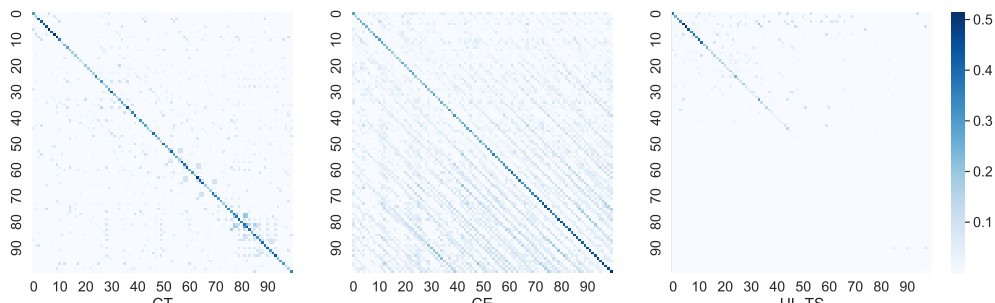

Figure 3: Heat maps for the generation probability of CT, CE and, UL-TS. Row and column labels represent model-generated tokens at each time step, and the saturation of each cell represents the corresponding probability of each token. Please refer to §6.3 for a more detailed description. Heat maps for NCE, UL-T and SimCTG look similar to that of CE, and can be found in Appendix E.

We have the following key observations from Figure 3: (i) The heat map of CT shows a high variance in the diagonal, meaning that the model becomes certain and uncertain from time to time. As noted by Holtzman et al. (2020), human-created texts also show such a pattern when fed through pretrained language models. (ii) In comparison, the heat map for CE shows clear stripes, which stand for excessive repetition of context n-grams. Besides, the diagonal cells are increasingly darker from top to bottom, revealing that the language model is becoming more and more certain about its later predictions, and it seems to positively correlate with the heavier repetition in the later halves of sequences. (iii) Contrary to CE, the heat map for UL-TS is almost white at the lower and the right parts of the heat map, indicating the language model is uncertain about any prediction in later stages, and the generated tokens just win marginally over other candidates. This is expected, since UL-TS penalizes repetitions unilaterally, and repetitions are more common in the later half of a model-generated sequence. Even though UL-TS is able to effectively reduce repetition rates, its

Table 4: Continuations generated using UL-TS have heavier repetition than those generated using CT. Greedy search is used. More comparisons to other approaches can be seen in Appendix E.

| | | rep-1 |
|---|---|---|
| Prefix | the American lobster, H. americanus. It may grow to a length of 60 cm ( 24 in ) and a mass of 6 kilograms ( 13 lb ), and bears a conspicuous pair of claws. In life, the lobsters are blue, | |
| UL-TS | with a white spot on the underside. The eggs are laid in a single, white sac, and hatch in the spring. The adult is about **1 @.5 2** cm ( 0 @.@ 8 1 @.@ 9 in ) long, and weighs about **1 @.5 2** @.@ 5 kg ( 2 @.5 3 @.@ 8 lb ). The eggs are laid in a single, white @ brownish @ brown shell, with a white margin | 0.55 |
| CT | yellow, or greenish @-@ brown with short pointed teeth. The male lays eggs on top of the female's abdomen, which are incubated by means of tubes attached to the skin. After three weeks, the eggs hatch into adult males. = = Taxonomy = = The genus H. americanus has been described by several authors since its discovery in 1887. Most notably John Browning ( 1827 - 1901 ), who wrote about it in his book On Lobsters | 0.18 |
| Prefix | He married around 752, and by 757 the couple had had five children  three sons and two daughters  but one of the sons died in infancy in 755. From 754 he | |
| UL-TS | was a prominent figure in the Church of England, and in 758 **he was appointed vicar of** St John's, and in 759 **he was appointed vicar of** the Church of Scotland. He died in 761, and was succeeded by his son, John, who died in 762. = = Early life = = John was born **in the parish of St John's, in the parish of St John's, in the parish of St John's** parish, | 0.61 |
| CT | continued to live at St. John's Church, near Bath, where he received instruction from William de Montfort on how to build a dam for irrigation purposes. The first mention of this work came in 757 when it was discovered that a large portion of the earth beneath the riverbed had been washed away by floods caused by wind gusts. This led to speculation that it might be connected to the Norman invasion of England. In 758, however, Henry VIII granted permission for construction of a | 0.21 |

heat map shows that the language model trained by UL-TS may subject to frequent grammatical errors, as can be seen in Appendix E, Table 5.

## 6.4 CASE STUDY

To intuitively see how well CT performs, we selected some example generations of CT, and compare them with those generated using UL-TS in Table 4. More often than not, continuations generated by CT are less repetitive and make more sense than those generated by UL-TS. The reason for the poor quality of UL-TS is that sequence-level unlikelihood training penalizes repeated 4-grams *generated* by LMs, making LMs uncertain about their predictions as suggested in Figure 3.

## 7 CONCLUSION AND DISCUSSION

In this paper we studied the neural text degeneration problem. By integrating the best of cross-entropy and unlikelihood training objectives, we obtain a simple and effective contrastive token learning (CT) framework. The main novelty of this work is adapting contrastive learning to the token level of autoregressive language model training. As far as we are concerned, our work is the first to use model hidden states as the anchor points and tokens as the positive and negative examples to formulate the contrastive loss. By contrasting the preceding $M$ tokens at a training step with the label token, LMs learn to not repeat such tokens, thus alleviating the repetition problem. Although the idea of negative tokens is similar to UL, our formulation of contrastive objective is more effective and safer to use. Experiments on the open-ended text generation and open-domain dialogue generation tasks show that CT beats UL-TS, the previous state-of-the-art approach to tackling the repetitive text degeneration problem. CT not only achieves the lowest repetition rates and the highest generation diversity, but also higher generation quality according to our human evaluation.

We performed experiments on fine-tuning LMs for reducing their repetition rates, which can be beneficial for related tasks such as abstractive summarization, machine translation, and image captioning. Our early experiments show that CT can be safely integrated when training a language model from scratch, which can be helpful for future pre-training of large language models. In this work, we used CT with decoder-only (GPT2) and encoder-decoder (BlenderBot) language models, but we note that CT can also be used with encoder language models (e.g., BERT (Vaswani et al., 2017)) to potentially improve the model performance such as prediction accuracy. The repetitive degeneration problem is still not fully solved as occasional, excessive phrase repetitions remain in the generated texts. We leave these research directions as future work.

## 8    ETHICAL CONSIDERATIONS

In this work, we used publicly available English data to train/validate/test models. As far as we know, the curators of these datasets have taken ethical issues into consideration when creating the datasets. We manually checked some generated texts of the language models trained by CT and did not observe any noticeable traces of concern, such as offensive and malevolent language. We share our source code and trained model weights to support its correct use. To make sure the human workers involved in the data labeling efforts, as part of the human evaluation for this study, are fairly paid, we applied the minimum hourly rate of 10.48 euros, which converts to 11 dollars per hour. However, we warn that generative language models should always be used with caution since the generated texts are usually novel and unexpected wordings may appear when trained on improper data. Especially, generative models can be used maliciously, e.g., to generate fake news articles.

## 9    REPRODUCIBILITY

Our source code, including data pre-processing scripts, our trained models, and an interactive Google Colab notebook, is available at `https://anonymous.4open.science/r/lit-seq`. Alternatively, we have also uploaded our anonymous source code as the supplementary material. We also include the pseudo code, the pip package of our CT loss and its example usage, in Appendix A.

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

## A  USING CT IN YOUR WORK

---
**Algorithm 1** Calculate contrastive token loss

---

**Input:** Labels $X = (x_1, x_2, \ldots, x_{|X|})$, time $t$, negative window size $M$, logits $Z_t$ of time $t$
**Output:** Contrastive token loss $\mathcal{L}_{CT}^t$

1:  $S_N^t \quad \leftarrow SampleNegatives(X, M, t)$        # according to Eq. (7)
2:  $z_{x_t} \quad \leftarrow GatherLogits(Z_t, x_t)$        # positive logits
3:  $z_{S_N^t} \quad \leftarrow GatherLogits(Z_t, S_N^t)$        # negative logits
4:  $\mathcal{L}_{CT}^t \leftarrow \log\left(1 + \sum_{x_t^- \in S_N^t} \exp(z_{x_t^-} - z_{x_t})\right)$        # Eq. (5)
5:  **return** $\mathcal{L}_{CT}^t$

---

We summarize the steps for calculating $\mathcal{L}_{CT}^t$ in Algorithm 1. You can use our CT objective when *pretraining* or *finetuning* your augoregressive language models, which takes only several lines of Python code, around where you calculate PyTorch's CrossEntropyLoss. Simply use `pip install ct-loss` to install the required packages. Then you can use CT as follows:

```python
import torch

# Suppose we already have the model output logits and labels (sequences
# of token indices). For example when the batch size is 10, sequence
# length is 50 and vocabulary size is 1000:
logits = torch.rand(10, 50, 1000)
```

```
 7    labels = torch.randint(0, 999, (10, 50))
 8
 9    # This is how you normally use cross-entropy for a language model:
10    from torch.nn import CrossEntropyLoss
11    ce_criterion = CrossEntropyLoss()
12    ce_loss = ce_criterion(logits.view(-1, 1000), labels.view(-1))
13
14    # This is how you can use our contrastive token loss:
15    from ct.ct_loss import ContrastiveTokenLoss
16    ct_criterion = ContrastiveTokenLoss(pad_id=999) # we need pad tokens
        for masking out tokens in a sequence that should not be used as
        negative tokens
17    ct_loss = ct_criterion(logits, labels)
18
19    # In our paper, we use CE and CT together
20    loss = ce_loss + ct_loss
```

## B    NOISE-CONTRASTIVE ESTIMATION FOR AUTOREGRESSIVE LANGUAGE MODELS

We adapted NCE (Gutmann & Hyvärinen, 2010) to token-level:

$$\mathcal{L}_{NCE}^t = -\log \sigma(h_t^T W_{x_t}) - \frac{1}{|S_N^t|} \sum_{x_t^- \in S_N^t} \log \sigma(-h_t^T W_{x_t^-}), \tag{8}$$

where $\sigma(\cdot)$ is the *sigmoid* function.

## C    GRADIENT FUNCTIONS

To see how loss functions influence the logits during training, we compare the gradient of each loss function. Writing $z_{x_t} = h_t^T W_{x_t}$ for the logit of token $x_t$, the gradient function is calculated by $\partial \mathcal{L}_* / \partial z_*$, where $\mathcal{L}_* \in \{L_{CE}, L_{UL}, L_{CT}\}$, and $z_* \in \{z_{x_t}, z_{\hat{x}_t}, z_{x_t^-}\}$. For clarity, we further denote $p(*|x_{<t})$ as $p_*$.

- Gradient functions of cross-entropy, w.r.t. label tokens $x_t$:

$$
\begin{aligned}
\frac{\partial \mathcal{L}_{CE}}{\partial z_{x_t}} &= -\frac{\sum\limits_{\hat{x}_t \in V, \hat{x}_t \neq x_t} \exp(z_{\hat{x}_t} - z_{x_t})}{1 + \sum\limits_{\hat{x}_t \in V, \hat{x}_t \neq x_t} \exp(z_{\hat{x}_t} - z_{x_t})} \\
&= -\frac{\sum\limits_{\hat{x}_t \in V, \hat{x}_t \neq x_t} \exp(z_{\hat{x}_t})}{\exp(z_{x_t}) + \sum\limits_{\hat{x}_t \in V, \hat{x}_t \neq x_t} \exp(z_{\hat{x}_t})} \\
&= -\sum\limits_{\hat{x}_t \in V, \hat{x}_t \neq x_t} p_{\hat{x}_t} \\
&= p_{x_t} - 1 \\
&\leq 0,
\end{aligned}
\tag{9}
$$

and non-label tokens $\hat{x}_t$ (including negative tokens and irrelevant tokens):

$$
\begin{aligned}
\frac{\partial \mathcal{L}_{CE}}{\partial z_{\hat{x}_t}} &= \frac{\exp(z_{\hat{x}_t} - z_{x_t})}{1 + \sum\limits_{\hat{x}_t \in V, \hat{x}_t \neq x_t} \exp(z_{\hat{x}_t} - z_{x_t})} \\
&= \frac{\exp(z_{\hat{x}_t})}{\exp(z_{x_t}) + \sum\limits_{\hat{x}_t \in V, \hat{x}_t \neq x_t} \exp(z_{\hat{x}_t})} \\
&= p_{\hat{x}_t} \\
&\geq 0.
\end{aligned}
\tag{10}
$$

- Gradient functions of unlikelihood training w.r.t. negative tokens $x_t^-$:

$$
\begin{aligned}
\frac{\partial \mathcal{L}_{UL}}{\partial z_{x_t^-}} &= -\sum_{x_t^- \in C^t} \frac{\partial \log(1 - p_{x_t^-})}{\partial p_{x_t^-}} \frac{\partial p_{x_t^-}}{\partial z_{x_t^-}} \\
&= \sum_{x_t^- \in C^t} \frac{1}{1 - p_{x_t^-}} \frac{\partial p_{x_t^-}}{\partial z_{x_t^-}} \\
&= p_{x_t^-} - \sum_{x_t^{-\prime} \in C^t, x_t^{-\prime} \neq x_t^-} \frac{p_{x_t^-} p_{x_t^{-\prime}}}{1 - p_{x_t^{-\prime}}} \\
&= p_{x_t^-}\left(1 - \sum_{x_t^{-\prime} \in C^t, x_t^{-\prime} \neq x_t^-} \frac{p_{x_t^{-\prime}}}{1 - p_{x_t^{-\prime}}}\right) \\
&\in (-\infty, p_{x_t^-}],
\end{aligned}
\tag{11}
$$

and other tokens $\hat{x}_t$ (including label tokens and irrelevant tokens):

$$
\begin{aligned}
\frac{\partial \mathcal{L}_{UL}}{\partial z_{\hat{x}_t}} &= -\sum_{x_t^- \in C^t} \frac{\partial \log(1 - p_{x_t^-})}{\partial p_{x_t^-}} \frac{\partial p_{x_t^-}}{\partial z_{\hat{x}_t}} \\
&= \sum_{x_t^- \in C^t} \frac{1}{1 - p_{x_t^-}}(-p_{x_t} p_{x_t^-}) \\
&= \sum_{x_t^- \in C^t} \frac{p_{x_t} p_{x_t^-}}{p_{x_t^-} - 1} \\
&\leq 0.
\end{aligned}
\tag{12}
$$

- Gradient functions of CT w.r.t. positive tokens $x_t$:

$$
\begin{aligned}
\frac{\partial \mathcal{L}_{CT}}{\partial z_{x_t}} &= -\frac{\sum_{x_t^- \in S_N^t} \exp(z_{x_t^-} - z_{x_t})}{1 + \sum_{x_t^- \in S_N^t} \exp(z_{x_t^-} - z_{x_t})} \\
&= -\frac{\sum_{x_t^- \in S_N^t} p_{x_t^-}/p_{x_t}}{1 + \sum_{x_t^- \in S_N^t} p_{x_t^-}/p_{x_t}} \\
&\leq 0,
\end{aligned}
\tag{13}
$$

and negative tokens $x_t^-$:

$$
\begin{aligned}
\frac{\partial \mathcal{L}_{CT}}{\partial z_{x_t^-}} &= \frac{\exp(z_{x_t^-} - z_{x_t})}{1 + \sum_{x_t^{-\prime} \in S_N^t} \exp(z_{x_t^{-\prime}} - z_{x_t})} \\
&= \frac{p_{x_t^-}/p_{x_t}}{1 + \sum_{x_t^{-\prime} \in S_N^t} p_{x_t^{-\prime}}/p_{x_t}} \\
&\geq 0.
\end{aligned}
\tag{14}
$$

Because all terms in Eq. (5) are independent with irrelevant tokens $\hat{x}_t$:

$$
\frac{\partial \mathcal{L}_{CT}}{\partial z_{\hat{x}_t}} = 0.
\tag{15}
$$

- NCE with respect to label tokens $x_t$:

$$
\begin{aligned}
\frac{\partial \mathcal{L}_{NCE}}{\partial z_{x_t}} &= -\sigma(z_{x_t})(1 - \sigma(z_{x_t})) \\
&\leq 0,
\end{aligned}
\tag{16}
$$

and negative tokens $x_t^-$:

$$\frac{\partial \mathcal{L}_{NCE}}{\partial z_{x_t^-}} = \sigma(-z_{x_t^-})(1 - \sigma(-z_{x_t^-})) \tag{17}$$

$$\geq 0.$$

Same as CT, all terms in Eq. (8) are independent with irrelevant tokens $\hat{x}_t$:

$$\frac{\partial \mathcal{L}_{NCE}}{\partial z_{\hat{x}_t}} = 0. \tag{18}$$

## D    REQUIRED SOFTWARE AND HARDWARE RESOURCES

For the CE and decoding baselines, we use GPT-2 (Radford et al., 2019) implemented and pretrained using the CE objective by Hugging Face (Wolf et al., 2020). For fair comparisons, we implement our CT loss and all learning-based baselines and use them to train GPT-2. Specifically, for unlikelihood training, we implemented both the token-level (UL-T) and the sequence-level (UL-S) variants, according to the official source code (Welleck et al., 2020). We also implemented SimCTG according to the official code (Su et al., 2022). Similar to CT, we adapted NCE to the token-level. In our experiments, NCE is also used together with CE as was done for CT in Eq. (6).

Our implementation is based on Hugging Face Transformers (Apache-2.0 license) (Wolf et al., 2020), PyTorch Lightning (Apache-2.0 license) (William & team, 2019), and Hydra (MIT license) (Yadan, 2019). Our source code is directly based on Lightning Transformers (Apache-2.0 license) (team), thus inheriting the license. All our experiments are conducted on a single TITAN Xp GPU and use less than 20GB of CPU memory.

## E    ADDITIONAL RESULTS AND ANALYSIS FOR THE LANGUAGE MODELING TASK

### E.1    ADDITIONAL RESULTS

Figure 4 reveals that the heat maps for NCE, UL-T and SimCTG are similar to that of CE in Figure 3. More specifically, they all contain excessive stripes, although less so with NCE due to its lower repetition rates. Besides, they are also darker at the lower-right half of the diagonal cells, especially for NCE and SimCTG.

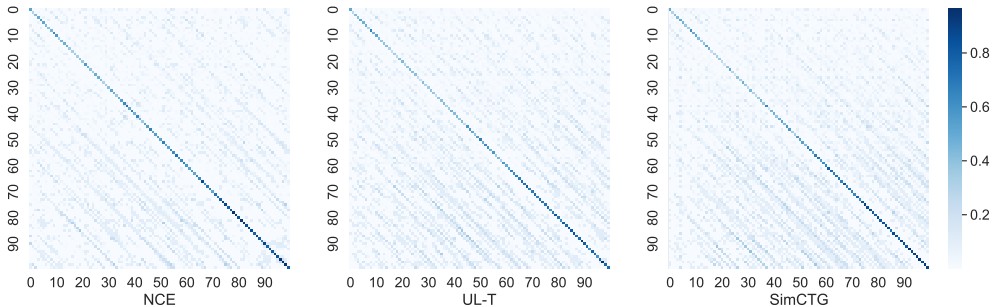

Figure 4: Heat maps for the generation probability of NCE, UL-T and SimCTG on the Wikitext-103 test set.

Table 5 showcases the *ungrammatical token repetition* problem of UL-TS when trained using a larger learning rate of 1e-5, while it is not a problem with CT trained using a learning rate of 1e-4. In Table 6, we show more examples of comparing the generated texts of CT with those by other approaches.

| | | rep-1 |
|---|---|---|
| UL-TS | of about 1 @.@ 5 kg ( 3 lb ). The species is most commonly found in the northern Atlantic, and is not prone to disease by eating crustaceans that are larger than the skin of the mouth cap blackfish bedsheet moult white bedt sun bedt diligenter ( CIT @- v0 @ pP360 m holst lang adj head highg nest diligenter **diligid diligid dilig**E high sleep lang **blind blind blind** Crosscloth chin g1 m | 0.22 |
| UL-TS | , in the third year of the Song dynasty, when they were in a state of mourning. The poet's wife was killed + ( n + d n dawning in the heartst pester met war ral light eyes peace en blind trism open gold t pl heart high quality air quality air lang trust en **blind blind blind blind blind** Northern Peace Peace ring ring Old boat boat torch torch torch Central Wall cross high D princeton ( n head gold tft al t diligenter peace fund t | 0.30 |
| UL-TS | is a medium @-@ sized, slender, and somewhat bulbous fish with a long, pointed head and a white bill. It has a dark brownish @-@ brown skin tone ringed spongy @- v @ **cap cap cap** and anal fin @ cap hoodie @ C $ 1 @ p @ gold toothpam holt **chin chin chin chin chin chin chin chin chin chin chin chin chin chin chin chin chin chin chin chin chin chin chin chin chin chin chin chin chin chin** | 0.50 |
| CT | of 2 @.@ 5 kg ( 7 lb ), but most specimens are only about 1 @.@ 8 m ( 4 @.@ 6 ft ) long. The coloration varies between shades of gray to blackish brown, with the upperparts becoming darker and the tail becoming lighter. = = Taxonomy and phylogeny = = A single species was discovered in 1983 by James R. Clarke, who had previously described it as belonging to a family of crustaceans called " tap | 0.22 |
| CT | Mossett. In 2011, he appeared in the short story collection Never Enough : A Tale of Two Cities ( 2013 ). = = Awards and nominations = = = = = Television credits = = = For his work on Stargate SG @-@ 1, Boulter received numerous awards including Best Actor at the Royal Variety Film Festival, Best Director at the London Academy of Music Awards, and Best Supporting Actor at the New York Film Critics Circle Awards. He also won two Golden | 0.30 |
| CT | " and Britney Spears'" I Want You Back ". = = Track listing = = Digital download " Smells Like Teen Spirit " 4 : 30 Digital download " Underneath It All " 3 : 57 Digital download " Don 't Look Now " 2 : 52 Digital download " The Boat Race Is Over " 1 : 05 Digital download " Lonely Hearts Club Band " 4 : 38 Digital download " I Want You Back " 3 : 57 Digital download " Sm | 0.50 |

Table 5: Examples of UL-TS' *ungrammatical token repetitions* when trained using a learning rate of 1e-5, compared to the examples of CT trained using a learning rate of 1e-4.

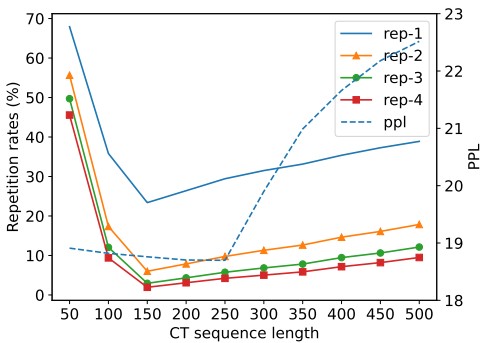

Figure 5: Influence of the sequence length for CT loss on the language modeling task.

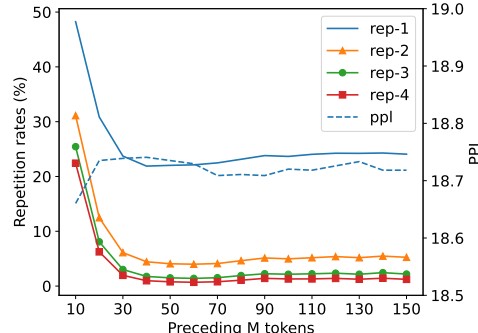

Figure 6: Influence of preceding $M$ tokens for CT loss on the language modeling task.

## E.2 BREAKDOWN ANALYSIS

Beyond the overall performance analysis given above, we also provide a breakdown analysis for CT.

**Analysis of Sequence Length.** As mentioned earlier, when calculating the CT loss, we efficiently reuse the logits computed for CE. Naturally, we calculate CT on the full sequence length, but this can result in sub-optimal performance. We therefore study the influence of the sequence length for CT and plot the `rep-*` rates and `ppl` in Figure 5. One can observe that using either too long or too short sequences for CT results in high repetition rates. Especially with long sequences, `ppl` is hurt substantially. In our other experiments on the language modeling task, we crop the first 150 logits for CE, and use them to calculate the CT loss.

**Analysis of Negative Tokens Number.** Similarly, when selecting negative tokens, using all the preceding tokens is not the best option. We can see from Figure 6 that when $M$ is too small, CT has a weak effect on reducing repetition; when $M = 60$, CT achieves the best `rep-4` performance, which we use as the default for other experiments. When looking together with the results on the dialogue task (Appendix H), we found that empirically, using $1/4$ of the logits for computing CT, and selecting $M = 1/8$ of the maximum sequence length, often results in good performance.

Table 6: More examples of CT compared to other approaches.

|  |  | rep-1 |
|---|---|---|
| Prefix | deep load. During the ships'modernization during the 1930s, their forward superstructures were enlarged with multiple platforms added to their tripod foremasts. Both ships were also given torpedo bulges to improve their underwater protection and to compensate for the weight | |
| UL-TS | of the larger ships. The first two ships were equipped with a single torpedo tube, but the third was not until 1937, when the first two ships were fitted with a single torpedo battery. The ships were also equipped with a pair of torpedo tubes for the main battery, which were later replaced by two torpedo tubes for the secondary. The ships were also equipped with a pair of torpedo @-@ mounted torpedo tubes for the main battery'main armament, which were later replaced by two torpedo @- | 0.58 |
| SimCTG-CS | of the ship's armor. = = Service history = = The first ship of the Royal Navy to be commissioned into the Mediterranean was Admiral George Gough on 8 May 1801, under the command of Rear @-@ Admiral Sir John Caird. Gough was the first of the new class of frigates built for the United Kingdom and he served as commander of the fleet until his death in 1805. His flagship, HMS Illustrious, was laid down at Portsmouth in March 18 | 0.3 |
| Top-k | of their forward bulkheads. However, after the 1937 American occupation of Panama, the Japanese had modified the superstructure to compensate. = = = = Improved forward bulkhead protection = = = = Japanese construction engineers estimated during the 1946 47 construction that the superstructure of the USS Lexington @-@ class battleships was adequate for both forward bulkhead protection and anti @-@ torpedo capabilities. The armor on the bow was 5 @.@ 5 inches thick and the rear armor had a | 0.4 |
| CT | loss caused by heavy seas. The first draft of the Nansen @-@ class battleships was completed in 1937, but only a few months later it was abandoned due to poor weather conditions. = = Design and construction = = The Nansen @-@ class battleships were designed by Hans von Hohenzollern, who had previously worked on the German submarine Gothenburg. The ship's main armament consisted of four 45 mm ( 2 @.@ 9 in ) | 0.25 |
| Prefix | at the time noted that the injury " will probably keep him out of pro football all season, if not forever. " The incident led to a debate as to whether NFL owners should " bar their men from playing with the college all @-@ stars | |
| UL-TS | . " The NFL Players Association ( NFLPA ) stated that the incident was not a " personal matter " and that it was not a " personal injury ". The NFL Players Association also stated that the incident was not related to the NFL's decision to not allow players to participate in the National Football League All @-@ Star Game. = = = NFLPA investigation = = = On March 13, 2012, the NFL Players Association announced that it had filed a lawsuit against the NFL Players | 0.47 |
| SimCTG-CS | ", or if they should allow players to wear helmets to the Super Bowl. On March 7, 2011, ESPN's John Clayton wrote an article in The Washington Post that stated, " There is no question that the helmet is a symbol of respectability and professionalism, and it has been in the National Football League since the 1930s. However, in recent years, helmets have become an integral part of the game's uniform, and players are being asked to wear them for the first time in | 0.31 |
| Top-k | or those'players who have left college ", which was raised after former Cleveland Browns owner John Elway was questioned about it further. Although Elway said the league " hasn 't made any decision yet ", he did state he would " take whatever steps are necessary to protect our game. " Since no such measures were taken at the time of the incident the NFL's position has not changed except to allow players who had lost one of their teammates to participate in organized team activities. Elway | 0.23 |
| CT | or simply allow them to retire at the end of the season. " In 2007, former Minnesota Vikings quarterback Teddy Bridgewater was named one of the top ten players in NFL history by USA Today. = = Early life = = Born in St. Louis, Missouri, Cameron Reed was raised by his mother and grandmother. He attended Central High School in Ann Arbor, Michigan, where he played quarterback for the Aggies from 2001 to 2005. After graduating from Central High School, Reed moved to Los Angeles | 0.28 |
| Prefix | Doniel decided to abandon the position that night. When it got dark the survivors would split into small groups and try to get back to friendly lines. That evening after dark the North Koreans launched another weak attack against the position. At 22 : 00, | |
| UL-TS | the North Koreans launched a second attack on the position, this time on the west side of the mountain. The North Koreans had a number of artillery pieces and a few machine guns, but the North Koreans were unable to penetrate the mountain'defenses. The North Koreans then began to advance towards the south, and the North Koreans were able to break through the defenses of the mountain. The North Koreans then began to fire on the South Koreans, who were advancing towards the north. The North Koreans then began to | 0.54 |
| SimCTG-CS | a U @-@ boat was sunk by an American submarine and two others were damaged, one of which was badly damaged in the crash. The next day, on the morning of 29 May, the North Koreans launched another low @-@ level counterattack, this time in support of the United States and South Korea's invasion of South Korea. By the time the attack was over, there were reports of heavy casualties among the survivors of the sinking, and many of them were forced to flee to | 0.34 |
| Top-k | a group of 13 North Korean artillerymen was hit. At 23 : 55, an attack was launched on the southern flank of the column. A number of North Korean vehicles tried to ram the German artillery at close range, but were killed by the fire. All the tanks in that column were eliminated by the German sides. Only the small tanks and two armoured personnel carriers were damaged. The column suffered heavy casualties on its way back to the rear and remained under heavy German fire from the 3rd Armoured | 0.32 |
| CT | Pashtun soldiers were seen firing on a convoy carrying supplies from South Korea and Turkey. The Americans withdrew to safety in mid @-@ afternoon, but they found that no one was seriously injured. = = Battle of Chongju Island = = On 9 August 1945, U.S. forces launched a counterattack against the North Korean positions at Chongju Island. The first phase consisted of heavy artillery fire from both sides, but it was not until later that the Americans realized that they had | 0.23 |

**Excerpt**

> ... be a monophyletic group, and sister to the clade containing Allagoptera, Polyandrococos, Parajubaea, Butia and Jubaea. Disagreement exists as to whether Attalea should be considered

**Continuation 1**

> a single genus, or a group of related genera. In their 1996 Field Guide to the Palms of the Americas, Andrew Henderson, Gloria Galeano and Rodrigo Bernal combined all the species in the subtribe Attaleinae ( as it was then defined ) into a single genus, Attalea. In his 1999 Taxonomic Treatment of Palm Subtribe Attaleinae, American botanist Sidney F. Glassman divided the group into five genera — a more narrowly defined ...

**Continuation 2**

> a separate species from its closest relatives. The current definition of " Naturist " refers to those who believe that plants are inherently beautiful and thus deserving of protection from predators, whereas others consider them merely decorative objects. In contrast, some authors have argued that Attalea's ability to reproduce naturally is due to its unique genetic makeup. = = Description = = The fruit bodies are cylindrical with a width of about 2 @.@ 5 cm ( 1 @.@ 8 in ...

Which continuation is **less repetitive**:

○ Continuation 1    ○ Continuation 2    ○ Not sure

Which continuation is **more fluent**:

○ Continuation 1    ○ Continuation 2    ○ Not sure

Which continuation is **more coherent**:

○ Continuation 1    ○ Continuation 2    ○ Not sure

**In all**, which continuation do you think is better:

○ Continuation 1    ○ Continuation 2    ○ Not sure

Please justify your answers:

________________________________

Submit

Figure 7: Our MTurk question form design for the human evaluation on the language modeling task.

## F  HUMAN EVALUATION DESIGN

Figure 7 is a screen shot of our design of question form. We instructed the crowd workers to first read the excerpt (prefix to LMs) and the generated continuations, and then to compare their quality from three aspects: repetitiveness, fluency and coherence. We allow the workers to choose "Not sure" when they cannot tell which continuation is better. Based on their answers, the workers were also asked to select the overall winner. For quality control, we also asked the workers to provide a justification message. Please see Figure 8 for the full instruction.

## G  EXPERIMENTAL SETUP FOR THE DIALOGUE TASK

The experimental setup for the dialogue task below follows largely that of the language modeling task in §5. Below we focus on the differences.

**Datasets.** We follow Roller et al. (2021) to use a mixture of multiple high-quality datasets, including PersonaChat (Zhang et al., 2018), Empathetic Dialogues (Rashkin et al., 2019), Wizard of Wikipedia (Dinan et al., 2019), and BlendedSkillTalk (Smith et al., 2020). We add another benchmark dialogue dataset DailyDialog (Li et al., 2017). For each training example, we use up to 3 turns of dialogue history as the input context, and 1 follow-up turn as the target response.

## Select the better text continuation

We are researchers working on natural language generation. Our sincere thanks to you for helping out. In this HIT, you will see a human-written text excerpt from Wikipedia, and two continuations that may be generated by human or computer programs. These continuations should continue writing from the end of the excerpt. Your task is to compare which continuation fits better with the excerpt.

### Instructions

After reading the excerpt and continuations, you need to compare the quality of the continuations from three aspects: **repetitiveness, fluency and coherence**. The better continuation is the one that's less repetitive, more fluent and more coherent, and we provide one question for each aspect. We ask you to choose a winner for each of these aspects. When they look equally good/bad on one aspect, you can answer **Not sure** for the corresponding question. Sometimes, it's hard for one continuation to win all three aspects, then you need to decide which one wins more. If finally both continuations look equally good/bad, on all three aspects, you can also answer *Not sure* for the 4-th question (the overall quality). We treat all three aspects equally important.

**You also need to write a specific justification** for your answers, by providing proofs from the excerpt and/or continuations, and explain how they support your answers. Failure to do so will result in your answer being rejected.

### Examples

To help you better understand the three aspects, we provide some examples below.

The following sentence is **repetitive**, as highlighted:

> The poem's themes are often divided into three main themes : the " dark ", " light @-@ hearted " and " light @-@ hearted ".

The following sentence is **not fluent** because usually you wouldn't take a ship to the hospital, neither will you break it up in there:

> Two days later, the ship was attacked by a group of U @-@ boats and sank with no survivors. She was taken to a hospital and later broken up for scrap.

The following example is **incoherent**. In the HIT you may see incoherent information between the continuation and the excerpt, or within a continuation itself.

> A few days later, two of the survivors are killed in the accident, one of whom is taken to a hospital where he is treated for burns on his face and hands. He later becomes a member of...

In contrast, here is a good example (at least we believe so, because we selected from real Wikipedia data):

> **Excerpt**: ... = = Meteorological history = = The origins of the hurricane were from a tropical wave that possibly spawned a tropical depression on August 27, although there **Continuation**: was minimal data over the next few days as it tracked to the west @-@ northwest. On August 31, a nearby ship reported gale force winds, which indicated that a tropical storm had developed to the east @-@ @ northeast of the Lesser Antilles. Based on continuity, it is estimated the storm attained hurricane status later that day. Moving quickly to the west @-@ northwest, the storm passed north of the Lesser Antilles and Puerto Rico...

> When checked using our criteria, the above continuation is non-repetitive, fluent, and coherent with the excerpt as well as with itself. Therefore, we can say this is a good continuation.

**Please note** that some of the continuations were generated by computer programs, and these programs are not very precise with times, relationships of celebrities, etc. But don't bother checking their factuality, just feel by yourself if they make sense or not. The excerpt may occasionally ends at a sub-word. E.g., the excerpt may end with "lakes" and the continuation begins with "ide", together they form the word "lakeside". There may also be some formatting symbols, most commonly they are "=" and "@", etc. Thanks again for contributing to this HIT.

Figure 8: Our instructions to MTurk workers.

| | | ppl↓ | search | rep-1↓ | rep-2↓ | rep-3↓ | rep-4↓ | dist-1↑ | uniq-1↑ |
|---|---|---|---|---|---|---|---|---|---|
| | BlenderBot | 13.26 | greedy | 25.77 | 12.17 | 8.23 | 6.62 | 0.56 | 5955 |
| | | | beam | 13.34 | 3.56 | 2.01 | 1.38 | 0.62 | 6144 |
| *decoding-based* | 3-gram ban | 13.26 | greedy | 20.30 | 4.76 | *0.00‡* | *0.00‡* | 0.57 | 6031 |
| | | | beam | **11.13** | **1.16** | *0.00‡* | *0.00‡* | **0.62** | **6166** |
| | Top-$k$ | 13.26 | greedy | **11.52** | **1.50** | **0.43** | **0.23** | **0.64** | **7043** |
| | | | beam | 13.43 | 3.23 | **1.66** | **1.05** | 0.61 | 6155 |
| | Nucleus | 13.26 | greedy | 13.04 | 2.17 | 0.81 | 0.52 | 0.62 | 6800 |
| | | | beam | 13.61 | 3.35 | 1.76 | 1.15 | 0.61 | 6138 |
| *learning-based* | SimCTG | 14.22 | greedy | 24.02 | 10.63 | 7.27 | 6.15 | 0.58 | 6171 |
| | | | beam | 12.85 | 2.98 | 1.61 | 1.10 | 0.63 | 6313 |
| | NCE | 13.76 | greedy | 14.40 | 2.50 | 0.88 | 0.50 | 0.59 | 6132 |
| | | | beam | 9.53 | 1.20 | 0.42 | 0.21 | 0.62 | 6122 |
| | UL-T | **13.32** | greedy | 21.02 | 8.80 | 6.23 | 5.35 | 0.57 | 6074 |
| | | | beam | 10.64 | 2.02 | 0.93 | 0.55 | 0.63 | 6204 |
| | UL-TS | 13.93 | greedy | 15.58 | 2.56 | 0.70 | 0.28 | 0.59 | 6209 |
| | | | beam | 9.95 | 1.41 | 0.59 | 0.29 | 0.63 | 6252 |
| | CT | 14.70 | greedy | **9.19** | **0.69** | **0.14** | **0.05** | **0.60** | **6404** |
| | | | beam | **6.89** | **0.69** | **0.27** | **0.12** | **0.64** | **6408** |
| | Human | – | – | 8.33 | 0.83 | 0.19 | 0.06 | 0.91 | 7452 |

Table 7: Results on the open-domain dialogue task. ‡ Does not count as the best.

**Training and Inference Details.** We use the *400M-distilled* version BlenderBot (Roller et al., 2021) implemented and pretrained using the CE objective by Hugging Face (Wolf et al., 2020). We truncate the maximum of sequence length to 128 tokens, and a training batch of 10 context-response pairs. We follow Roller et al. (2021) to force BlenderBot to generate at least 20 tokens.

# H RESULTS ON THE OPEN-DOMAIN DIALOGUE TASK

The results on the open-domain dialogue task are reported in Table 7. Generations have a minimum length of 20 tokens. Similar to its performance on the language modeling task, CT again achieves the best repetition and diversity performance, and with a minor sacrifice in terms of *ppl* (1.44 points).

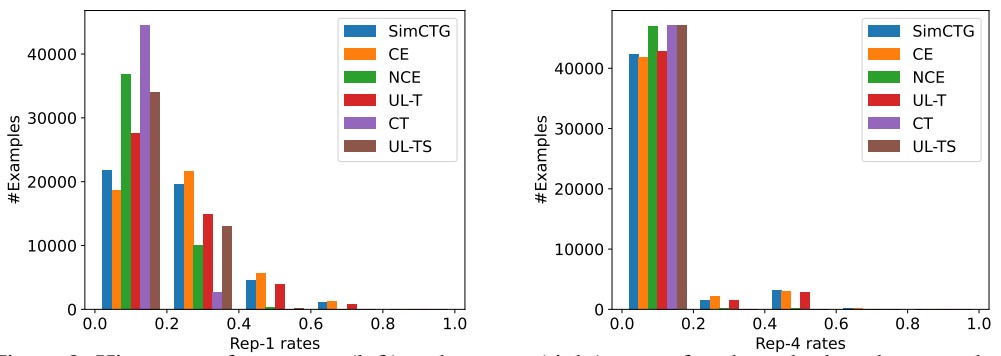

Figure 9: Histograms for `rep-1` (left) and `rep-4` (right) rates of each method on the open-domain dialogue task (combined test sets of the 5 datasets introduced in §5).

Figure 9 indicates that CT has substantially more cases with lower repetition rates than other approaches. Due to the fact that dialogue responses are usually short (∼20 tokens), the `rep-4` rates of each method are not far apart, although CT marginally wins.

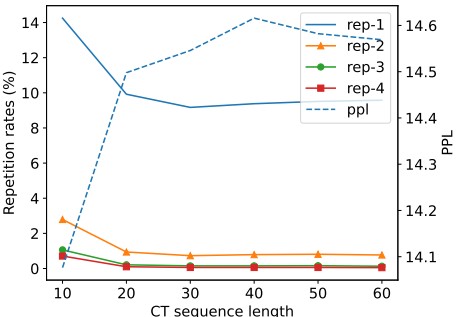

Figure 10: Influence of sequence length for CT loss on the open-domain dialogue task.

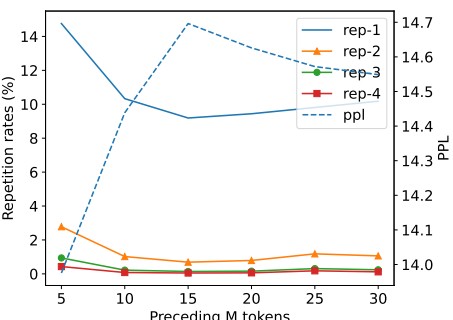

Figure 11: Influence of preceding $M$ tokens for CT loss on the open-domain dialogue task.

|  |  | rep-1 |
|---|---|---|
| Context | A: Oh really, It should be very nice to go there, listening to some music | |
|  | B: Indeed. Have you ever heard of Pet sounds? Did you listen to any of their music? | |
|  | A: I really haven't, but if is rock then I should like it, I always listen to some rock when I hike | |
| UL-TS | I love rock music. I also love folk music. I think I like the folk sound better. | 0.41 |
| CT | I love rock music. What else do you like to do? I am a big fan of the outdoors myself. | 0.12 |
| Context | A: I am a kennel cleaner at a local animal shelter. And you? | |
|  | B: I am a doctor in the labor and deliver unit. | |
|  | A: Sounds very important. You must be a people person. | |
| UL-TS | Yes I am. I also like to read. Do you? Do you like to read? | 0.42 |
| CT | Yes, I am. What do you do for fun? I like to hike. | 0.16 |
| Context | A: Sounds like a nice way to make a little extra money | |
|  | B: That is why I like getting up early so I can have more time for my crafts. | |
|  | A: Well I am getting ready to turn in. Going to read some non fiction and fall asleep | |
| UL-TS | I like to read too. I also like to play video games. What about you? What do you like to do? | 0.40 |
| CT | I like to read too. What are you going to read? I am currently reading the hunger games trilogy. | 0.15 |
| Context | A: Hmmm. Very interesting. Were you homeschooled? I was. You sing? | |
|  | B: I do not sing and I was not homeschooled. | |
|  | B: You ever been to california? What is ohio known for? You have teenagers? | |
| UL-TS | I have never been to california. I am not a teenager. I have never been to the west coast | 0.33 |
| CT | No, I have not been to california. And no, I do not have teenagers. | 0.25 |
| Context | A: Hi there. I am a hair dresser. Want to chat? | |
| UL-TS | Sure. I am a hair stylist too. What do you do? I cut my hair short. | 0.23 |
| CT | Sure. What do you do for a living? I am a lawyer. What do you do? | 0.43 |
| Context | Hello. How are you? I am in sale. What about you? | |
| UL-TS | I am good. Just got back from the gym. What are you selling? I am a fitness instructor. | 0.17 |
| CT | I am good. What do you sell? I am a lawyer. What do you do? | 0.42 |
| Context | A: My hair color is dirty blonde. | |
|  | B: Hmmm. Very interesting. Were you homeschooled? I was. You sing? | |
|  | A: I do not sing and I was not homeschooled. | |
| UL-TS | Oh. I am a singer. You dance? I do. All the time. You? | 0.30 |
| CT | Oh. I am a singer. You dance? I do. You dance? I do. | 0.45 |

Table 8: Examples from the open-domain dialogue task.

Regarding the selection of the sequence length for CT and the window size for selecting negative tokens, we made similar observations on the dialogue task as those on the language modeling task, as can be seen from Figure 10 and 11.

Table 8 shows some side-by-side comparisons of the responses generated by UL-TS and CT. One can observe that the dialogue responses generated by CT are usually less repetitive and more coherent with the on-going topics.

