# OpenReview forum: "A Simple Contrastive Learning Objective for Alleviating Neural Text Degeneration"
_ICLR.cc/2023/Conference — Submitted to ICLR 2023_

### Official Review · Reviewer_YrBU · 2022-10-23

**Confidence:** 4
**Correctness:** 3
**Technical Novelty And Significance:** 2
**Empirical Novelty And Significance:** 2
**Recommendation:** 5

**Clarity, Quality, Novelty And Reproducibility:**

- reproducibility: good - provides code samples, google colab (https://anonymous.4open.science/r/lit-seq)  and pip-package
- "Unlikelihood training also unintentionally boosts the probability of other irrelevant tokens." -- can you explain more?


**Strength And Weaknesses:**

## Strengths
- A new training objective that is well-motivated
- both automatic and human evaluation
- Interesting view of cross-entropy in the lens of contrastive learning.
- well-written

## Weakness
- For negative tokens, authors use previous M (section 3.2), whereas UL uses all previous. It is unclear how important this is. One way to measure it is to compare performance using all tokens as in UL.
- repetition seems to be less of an issue with better models (larger and/or trained with more data), so it is unclear how useful this technique is for the best models. Experiments were done with GPT-2 small. How do results change with larger GPT2 models?
- The human eval (Table 3) does not show any statistical significance compared to UL-TS, one of the baselines. In particular, some claims about UL-TS being less fluent (ungrammatical repetitions) does not agree with the human eval.


**Summary Of The Paper:**

Proposes a new training objective, contrastive token learning (CT), for language models that contrasts the target token with M previous tokens (as negatives). It is compared to regular cross-entropy and other baselines, and especially unlikelihood training (UL). Similar to UL it is motivated by the observation that CE training can lead to degenerate generation in the form of repetition, and it is shown that the proposed objective can significantly reduce it while keeping perplexity similar to CE.

One key idea is categorizing tokens as positive (label/target), negative (repeated), and irrelevant (others). Authors believe negative and irrelevant tokens should not be treated the same as in CE, and instead only penalize negative tokens in the contrastive oss.

The LM is evaluated on the wikitext-103 task and compared primarily with CE and UL. A human evaluation 1 vs 1 shows that CT is preferred to baselines, although it is not statistically significant .


**Summary Of The Review:**

While the proposal of CT as a new training objective is interesting, it is unclear how important the effect of using M previous tokens instead of all tokens as UL does. A comparison of (a) with CT using all previous tokens; or (b) UL with only M previous tokens would help understand the effect. Furthermore, it is unclear how important the repetition issue is for better/larger models, as these typically do not use any repetition-reducing heuristics and seem to suffer less degenerative generation; some scaling experiments could help resolve this question. Finally human evaluation doesn't seem to prefer CT over UL-TS in-spite of claims of better fluency/repetition.

In conclusion, while the paper is well-written, I don't yet see (without more experiments) the utility of the proposed technique over standard cross-entropy training.

---

### Official Review · Reviewer_fiLs · 2022-10-25

**Confidence:** 4
**Correctness:** 3
**Technical Novelty And Significance:** 1
**Empirical Novelty And Significance:** 2
**Recommendation:** 3

**Clarity, Quality, Novelty And Reproducibility:**

The paper is very easy to understand and follow, but it lacks novelty.


**Strength And Weaknesses:**

Strength:
This paper has a very intuitive idea, and it is easy to follow.
Experimental results and human evaluations are positive.


Weaknesses:
There is not enough novelty in the paper.



**Summary Of The Paper:**

The paper proposes a new contrastive token (CT) learning objective to teach a LM to generate high probabilities for label tokens and low probabilities for negative candidates (repetitive tokens). The idea of this paper is very similar to "unlikelihood training", however, as the authors showed in section 3.3, in unlikelihood training the irrelevant tokens are promoted, while in CT they remain unchanged, and the negative tokens are sometimes promoted and sometimes suppressed by the gradient function in unlikelihood training, while they are always suppressed in CT.

**Summary Of The Review:**

There is not enough contribution in the paper.

---

### Official Review · Reviewer_8fJN · 2022-10-29

**Confidence:** 4
**Clarity, Quality, Novelty And Reproducibility:** 1. The paper is very clearly written …
**Correctness:** 3
**Technical Novelty And Significance:** 3
**Empirical Novelty And Significance:** 3
**Recommendation:** 5

**Strength And Weaknesses:**

Strengths:
1. The proposed loss has the sound motivation and is very simple to implement and can be added into any generation system with little effort.

2. The results show improvement on many metrics across both tasks.

Weaknesses:
1. The experimental setup is weak. The primary result provided in the main paper is on GPT2-small fine-tuned on wikitext which by today's LM standards is not a convincing setup. Further experiments need to be conducted on larger versions of GPT2 or models trained with more data same as prior work. Furthermore, the metrics used and how they are interpreted deviate from other works. In this work, perplexity is used as a criterion for quality (lower the better). Prior work (https://arxiv.org/abs/1904.09751) has shown that low perplexity in GPT2 based models is not a good indicator of quality but rather of degenerate behaviour and closeness to human text perplexity is a better measure. Additionally, most prior works report dist-2,3 and even 4 but this works only reports dist-1. Also by the definition of dist-1 it should always be less than 1 since it measures the fraction of unique n-grams in the output text. I'm confused by the values reported in the paper.

2. Also unclear is the meaning of greedy or beam search in top-k (or p) sampling. Further,  it is not clear why beam search is the choice of decoding algorithm. General LMs like GPT2 (or their fine-tuned versions) and even dialogue models in principle have multiple potential continuations given a prompt while beam search gives out just one output.

3. Some crucial baselines like simple ascetral sampling (that top-k with k=vocab size) and more recently proposed typical sampling (https://arxiv.org/abs/2202.00666).

**Summary Of The Paper:**

This paper presents a new loss function to train text generation models aimed at reducing repetitions commonly associated with models trained with a simple cross-entropy loss. Interpreting cross-entropy as a contrastive loss function, this work proposes to add to this loss another term similar to cross entropy which considers M tokens generated before every step at negative examples. The primary motivation behind this addition is that cross entropy treats negative tokens and irrelevant tokens equally which this work aims to fix. On experiments conducted on a fine-tuned GPT2-small and a dialogue generation system, the authors claim increased quality (by perplexity), diversity (by dist-1 and uniq-1) and reduced repeated phrases.

**Summary Of The Review:**

While the method has an interesting motivation, the experimental setup seems small and not very convincing both in terms of LMs considered and the metrics reported. I am currently leaning negative but would be willing to revise my score after rebuttal.

---

### Official Review · Reviewer_TkXN · 2022-10-30

**Confidence:** 4
**Correctness:** 3
**Technical Novelty And Significance:** 2
**Empirical Novelty And Significance:** 2
**Recommendation:** 3

**Clarity, Quality, Novelty And Reproducibility:**

**Clarity**
This paper is very clear.

**Quality**
This paper does not meet the bar of ICLR. The method is not novel and the experiments are pretty limited.

**Novelty**
Thin. Unlikelihood learning has made the most contributions.

**Reproducibility**
Good. The authors have uploaded the code.

**Strength And Weaknesses:**

**Strength**
1. The paper is easy to follow and the idea is intuitive.
2. Several case studies are given which are encouraged.

**Weaknesses**
1. The novelty is rather thin. This paper is an incremental work on UL: 1) The core idea that penalizes the previously generated tokens has been proposed by UL; 2) I **totally disagree** with the author's claim that `Comparing Eq.(5) to Eq. (4), we see that UL only
considers the probabilities of negative tokens`, Equation 4 of the UL paper clearly shows that UL has jointly considered the probabilities of positive and negative tokens (i.e., a likelihood term for a positive token and an unlikelihood term for negative tokens). However, the authors try to hide this important detail and only write the likelihood term of negative tokens in Equation 4 of the current paper.
2. The experiment is insufficient. The authors have cited many machine translation papers and also state that `We performed experiments on fine-tuning LMs for reducing their repetition rates, which can be beneficial for related tasks such as abstractive summarization, machine translation, and image captioning.` Therefore, why not simply perform the proposed method in these tasks? PPL is not a reliable metric for evaluating text generation tasks.  The metrics (especially the recently proposed neural metrics ) in abstractive summarization, machine translation, and image captioning are more competent to give a more reasonable evaluation, which can make the proposed method more convincing. A small suggestion: if you think your proposed method is simple and still can be accepted by a top-tier conference, the method has to be very powerful or very general. Unfortunately, the proposed method does not show its effectiveness.

**Summary Of The Paper:**

This paper proposes a contrastive learning method to balance the learning of positive and negative tokens in text generation tasks (e.g., language modeling and open-domain dialogue generation tasks).

**Summary Of The Review:**

Given the thin novelty and insufficient experiments, I suggest rejecting this paper.

---

### Decision · Program_Chairs · 2023-01-20

**Decision:**

Reject

**Justification For Why Not Higher Score:**

The theoretic and empirical contributions of the approach are not strong enough to justify acceptance.

**Justification For Why Not Lower Score:**

N/A

**Metareview: Summary, Strengths And Weaknesses:**

The paper proposes a contrastive token training objective for autoregressive language modelling. The results on WikiText-103 LM generation show that, on repetition metrics, the method performs better than previous learning-based methods and on par with the best previous decoding-based method (SimCTG-CS). On human evaluation the method also performs better than previous approaches on most criteria. While the paper provides a valuable result, it is an incremental modification of unlikelihood training, and the experimental setup is not strong enough: related methods have been applied on multiple downstream tasks, the automatic metrics used are not sufficient to evaluate generation quality, and the human evaluation is not statistically significantly better than UL-TS.